



# Estimating freshwater flux amplification with ocean tracers via linear response theory

Aurora Basinski-Ferris[1] and Laure Zanna[1]

[1]Courant Institute of Mathematical Sciences, New York University

**Correspondence:** Aurora Basinski-Ferris (abf376@nyu.edu)

**Abstract.** Accurate estimation of changes in the global hydrological cycle over the historical record is important for model evaluation and understanding future trends. Freshwater flux trends cannot be accurately measured directly, so quantification of change often relies on trends in ocean salinity. However, anthropogenic forcing has also induced ocean transport change, which imprints on salinity. We find that this ocean transport affects the surface salinity of the saltiest regions (the subtropics), while having little impact on the surface salinity in other parts of the globe. We present a method based on linear response theory which accounts for the regional impact of ocean circulation changes while estimating freshwater fluxes from ocean tracers. Testing on data from the Community Earth System Model large ensemble, we find that our method can recover the true amplification of freshwater fluxes, given thresholded statistical significance values for salinity trends. We apply the method to observations and conclude that over the period 1975 to 2019, the hydrological cycle has amplified by $4.52 \pm 1.21\%$ per degree of surface warming.

## 1 Introduction

Under anthropogenic forcing, the hydrological cycle is expected to intensify; both global mean precipitation and the pattern of evaporation minus precipitation (E-P) are predicted to amplify (Hegerl et al., 2015). Global mean precipitation is expected to increase at a rate of $2-3\%$ $^o$C$^{-1}$, less than the Clausius-Clapeyron rate ($7\%$ $^o$C$^{-1}$), due to energetic constraints governing tropospheric radiative cooling (Chou and Neelin, 2004; Muller and O'Gorman, 2011; Trenberth, 2011; O'Gorman, 2012; Allan et al., 2014). This $2-3\%$ $^o$C$^{-1}$ rate is also predicted by climate models – e.g., the CMIP6 ensemble projects an increase in mean precipitation of $2.1-3.1\%$ $^o$C$^{-1}$ in abrupt 4 times CO$_2$ experiments (Pendergrass, 2020). However, the amplification rate of E-P patterns, such that wet regions get wetter and dry regions get drier, is more uncertain. While this pattern amplification is linked to the Clausius-Clapeyron relationship, changes in the large scale atmospheric circulation could result in a smaller intensification rate (Held and Soden, 2006), therefore limiting theoretical predictions of the amplification of the E-P pattern. Recent generations of the CMIP model ensembles predict amplification values less than Clausius-Clapeyron, but with large spread. The CMIP5 model ensemble projects an amplification of the E-P pattern of $4.3 \pm 2.0\%$ $^o$C$^{-1}$ (Skliris et al., 2016). In CMIP6, E-P pattern intensification is estimated at $0.3-4.6\%$, based on freshwater transport from the subtropics and tropics to polar regions over the period 1970 to 2014 (Sohail et al., 2022).



Given the gap in theoretical understanding and the spread of amplification in models, an accurate benchmark of E-P pattern change over the historical period is needed. However, due to the difficulty in measuring freshwater flux trends directly, changes both in mean precipitation and the E-P pattern are uncertain over the historical record (Hegerl et al., 2015). Detection of amplified E-P patterns are mostly approached using ocean salinity changes (Douville et al., 2021). This idea of using salinity as a "rain gauge" stems from Wust (1936) who highlighted the similarity between surface salinity and E-P patterns. Recently, studies have used either surface salinity patterns (Durack et al., 2012; Zika et al., 2018) or three dimensional ocean salinity (e.g., Hosoda et al., 2009; Curry et al., 2003; Helm et al., 2010; Skliris et al., 2014; Cheng et al., 2020; Sohail et al., 2022) to infer changes in surface freshwater fluxes. For the rest of this paper, we refer to the amplification of the E-P pattern (rather than changes in mean precipitation) when referencing change in the hydrological cycle.

Estimates of hydrological cycle change using surface salinity leverage that amplification of the E-P pattern is expected to contribute to salinity pattern amplification whereby areas of the ocean surface that are saltier (fresher) than average in the climatological mean get saltier (fresher) (Durack et al., 2012). The AR6 report states that the near-surface salinity contrast between high and low salinity regions has increased by 0.07-0.20 pss (practical salinity scale) from 1950 to 2019 (Fox-Kemper et al., 2021). Durack et al. (2012) estimated amplification of freshwater fluxes by defining a pattern amplification metric as the linear regression between climatological values of a quantity (e.g., surface salinity) and trends of that quantity at the same location. The relationship between the pattern amplification of salinity and of E-P in CMIP3 models was applied to infer E-P changes from surface salinity observations. This empirical relationship between changes in salinity and E-P patterns in models is one way estimate fluxes from ocean tracers. However, the local change in surface salinity can be affected both by freshwater fluxes and by changes in ocean transport (e.g., Zika et al., 2018). Vinogradova and Ponte (2017) highlighted that local surface salinity is affected by non-negligible effects of isopycnal mixing, diapycnal mixing, and advection. Turner et al. (2022) estimated redistributed salinity, where a redistributed tracer field is defined as the reorganization of the pre-industrial tracer through anomalous transport. They found redistributed salinity was significant at the surface in some regions, although still smaller than the effect of E-P. Additionally, Sohail et al. (2022) showed that in some regions of the ocean (e.g., the 2% warmest volume), salinity changes are influenced by both surface fluxes and ocean transport. Finally, recent work shows significant heat redistribution over the historical record (e.g. Winton et al., 2013; Gregory et al., 2016; Bronselaer and Zanna, 2020). These changes in ocean transport suggest an impact of ocean dynamics on local salinity trends. Zika et al. (2018) estimated hydrological cycle amplification utilizing the surface pattern amplification metric from Durack et al. (2012) and accounting for the impact of an idealized surface heat flux on the salinity pattern. Thus, this work accounted for heat flux induced ocean transport changes under the assumption that this redistribution contributes to a linear amplification of the climatological salinity pattern.

Studies linking interior ocean salinity change to surface freshwater fluxes have either examined salinity changes in volumes bounded by depth surfaces (Hosoda et al., 2009; Curry et al., 2003; Skliris et al., 2014; Cheng et al., 2020) or in watermass-based frameworks (Helm et al., 2010; Skliris et al., 2016; Zika et al., 2015; Sohail et al., 2022). Using full depth salinity negates difficulties with discerning between local changes due to the imprint of fluxes versus ocean transport. However, this approach





is limited by reliance on interior salinity data which has larger uncertainties than the surface due to poor sampling (Durack,
2015; Cheng et al., 2020).

In this work, we present a novel estimate of E-P pattern amplification over the historical period. We hypothesize (and will
test in this paper) that ocean transport change induced by heat fluxes contributes to different regional changes in salinity. We
propose a methodology which can account for these local effects of ocean transport when estimating fluxes from tracers. Here,
we focus on surface salinity, which is more reliable than full depth salinity, and on the application of linear response theory
(Ruelle, 2009) regionally, therefore relaxing assumptions made in Durack et al. (2012) and Zika et al. (2018). Our method
relies on two assumptions, which we will validate: 1) the change in regional ocean tracers under anthropogenic forcing can be
expressed as the sum of the effect from freshwater fluxes, heat fluxes, and wind stress change applied separately; 2) for each
component (e.g., freshwater flux forcing), the response of surface tracers is linear with respect to the forcing and therefore
linear response theory can be applied.

In this paper, we first introduce the main methodology, based on a specific application of linear response theory, in Sect. 2.
We then validate and test the method using data from the Community Earth System Model (CESM) in Sect. 3. In Sect. 4, we
apply the method to observations and estimate freshwater flux changes over the period 1975 to 2019.

## 2    Methodology

We propose a method to estimate freshwater fluxes from surface salinity pattern change, while accounting for the impact of
heat flux or wind stress induced circulation change on salinity. The method is based on a specific application of linear response
theory on a few key surface regions. These surface regions are defined as locations with similar salinity, as identified using a
Gaussian Mixture Model (an unsupervised machine learning algorithm).

### 2.1    Linear response theory

We start by reviewing linear response theory. Response theory, a generalization of the fluctuation dissipation theorem, can be
applied to non-equilibrium and chaotic dynamical systems, such as climate, to predict statistical properties both near and far
from equilibrium (Lucarini and Sarno, 2011; Lembo et al., 2020). Following other climate studies (Lucarini and Sarno, 2011;
Lembo et al., 2020; Ragone et al., 2016; Torres Mendonça et al., 2021), we use linear response theory from Ruelle (2009)
which assumes a dynamical system of the form

$$\frac{d\mathbf{x}}{dt} = \mathbf{A}(\mathbf{x}) + \mathbf{B}(\mathbf{x})f(t), \tag{1}$$

where $\mathbf{x}(t)$ is the state vector (e.g., a possibly infinite vector encoding the state of the climate system at time $t$), $\mathbf{A}(\mathbf{x})$ is a
vector field representing the system's unperturbed dynamics, $\mathbf{B}(\mathbf{x})$ is a vector field representing the pattern of the forcing, and
$f(t)$ is the time evolution of the forcing (e.g., anthropogenic emissions).





Consider a given vector of observables, $\mathbf{Y}(\mathbf{x}(t))$, which are quantities observed or measured that depend on the state vector of the system. The expectation value of $\mathbf{Y}(\mathbf{x}(t))$ can be written as

$$\langle \mathbf{Y} \rangle (t) = \langle \mathbf{Y} \rangle_0 + \sum_{n=1}^{\infty} \langle \mathbf{Y} \rangle^{(n)}(t), \tag{2}$$

where $\langle \mathbf{Y} \rangle_0$ is the expectation value in the unperturbed state and $\langle \mathbf{Y} \rangle^{(n)}(t)$ gives the $n$-th order perturbative contribution.

Considering only the first order (linear) contribution, the expectation of $\Delta \mathbf{Y}(t)$, the deviation of the observable vector from the unperturbed system, is given by

$$\langle \Delta \mathbf{Y} \rangle (t) = \int_0^t \boldsymbol{\chi}(t-\tau) f(\tau) d\tau + O(f^2), \tag{3}$$

where $\boldsymbol{\chi}(t)$ is a vector of the Green's functions of the observables. Thus, $\boldsymbol{\chi}(t)$ is the response of the system to a Dirac delta function in the same variable as the forcing $f(t)$; if $f(t)$ is anthropogenic $CO_2$ emissions, then $\boldsymbol{\chi}(t)$ is the response to an instantaneous impulse of $CO_2$. Equation (3) can also be written as

$$\langle \Delta \mathbf{Y} \rangle (t) = \int_0^t \mathbf{R}(t-\tau) \frac{df}{d\tau}(\tau) d\tau + O(f^2), \tag{4}$$

where $\mathbf{R}(t)$ is a vector of the observables' responses to a step forcing rather than an impulse forcing (Hasselmann et al., 1993).

## 2.2 Estimating surface fluxes using linear response theory

In this section, we outline our methodology to infer surface fluxes from regional ocean tracer data using an application of response theory.

### 2.2.1 Dimensionality reduction to characterize salinity pattern

We aim to relate global surface fluxes to the evolving surface salinity pattern. We track pattern evolution using trends in regional salinity, and thus, first find a reduced number of regions that make up the salinity pattern. This dimensionality reduction is performed using a Gaussian Mixture Model (GMM) which fits a distribution as the sum of $n$ Gaussians (called mixtures), each with its own mean, standard deviation, and weight. The weights of all the $n$ mixtures sum to 1 (Brunton and Kutz, 2019). Here, we fit a GMM to the climatological surface salinity distribution to decompose the global salinity pattern into regions of similar

salinity. Details on the application to data including the period used to compute climatologies is in Sect. 3. In practice, we fit the GMM using the sklearn package in Python, which implements the Expectation-Maximization algorithm (Pedregosa et al., 2011). We make paramater choices of 40 initial conditions and a convergence tolerance of $10^{-3}$. After fitting, we categorize each point on the ocean surface into a mixture if the climatological salinity of that point is most likely to lie in that particular Gaussian.

GMMs are useful for identifying patterns in oceanographic data (e.g. watermass identification) and can be used as an intermediate between local analysis and domain averaging (Maze et al., 2017; Jones et al., 2019). However, GMMs are sensitive to





the number of mixtures; this choice can be guided by criteria such as the Akaike information criterion (AIC) and the Bayesian information criterion (BIC) to balance overfitting with goodness of fit, but also depends on the scientific question being addressed. We explore the sensitivity to mixture number for our application in Sect. 4.

The dimensionality reduction step allows us to reduce the complexity of the problem as we can characterize the change in the salinity pattern by considering the trends in a few ocean surface regions. We also have avoided imposing that the evolving pattern is a scaled version of the climatological pattern, as each region making up the surface pattern can have a different trend.

### 2.2.2  Flux estimation using the ensemble mean

As shown in Eq. (4), linear response theory requires the response of observables to a step forcing, $\mathbf{R}(t)$. In this proposed
framework, the observables are ocean tracers in surface regions found by the GMM in the dimensionality reduction step (see Sect. 2.2.1). Although we expect freshwater fluxes to have the strongest imprint on salinity, we utilize surface temperature as a tracer in the method as an additional constraint.

Here, we are aiming to separate out the effect of heat flux, freshwater flux, and wind stress change. Thus, we utilize the assumption that the change in regional ocean tracers under anthropogenic forcing can be written as the sum of the effect due
to individual forcing components imposed separately. We apply linear response theory from Eq. (4) to each individual forcing component, resulting in

$$\langle \Delta \mathbf{Y} \rangle(t) = \int_0^t \mathbf{R}^h(t-\tau)\frac{dF^h}{d\tau}(\tau)d\tau + \int_0^t \mathbf{R}^w(t-\tau)\frac{dF^w}{d\tau}(\tau)d\tau + \int_0^t \mathbf{R}^s(t-\tau)\frac{dF^s}{d\tau}(\tau)d\tau. \tag{5}$$

In Eq. (5), $\langle \Delta \mathbf{Y} \rangle(t)$ has $2n$ rows, where each row is a time series of surface salinity and temperature in one of the $n$ mixtures. Temperature and salinity change are normalized in the $L^2$ norm such that the scale of change of temperature observables is equal to the scale of change of salinity observables. The tracers are then area weighted using the area of the GMM region to
which they belong. $\mathbf{R}^h(t)$, $\mathbf{R}^w(t)$, and $\mathbf{R}^s(t)$ are also each composed of $2n$ rows. Each row is a time series of (normalized and area weighted) temperature and salinity responses to heat flux, freshwater flux, and wind stress step forcings respectively in each GMM region. Finally, the terms $\frac{dF^h}{dt}$, $\frac{dF^w}{dt}$, and $\frac{dF^s}{dt}$ are derivatives of the time evolutions of each forcing as a proportion of the strength of the step forcings used for the response functions $\mathbf{R}^h(t)$, $\mathbf{R}^w(t)$, and $\mathbf{R}^s(t)$. For example, if the heat flux response at time $t_1$ was attributed to a forcing exactly equal to the step forcing, then $F^h(t_1) = 1$.
The system in Eq. (5) is discretized with a time step, $\Delta t$, equal to one year. We choose this time discretization as it is the shortest time step that smooths over seasonal variability in temperature and salinity data. Thus, the discretized version of Eq. (5) is

$$\langle \Delta \mathbf{Y} \rangle(t) = \sum_{k=0}^m \mathbf{R}^h(m-k)\frac{dF^h}{dt}(k) + \sum_{k=0}^m \mathbf{R}^w(m-k)\frac{dF^w}{dt}(k) + \sum_{k=0}^m \mathbf{R}^s(m-k)\frac{dF^s}{dt}(k). \tag{6}$$

Here, the known variables are the response functions and the ensemble averaged observable vector anomaly, $\langle \Delta \mathbf{Y} \rangle(t)$. At each time step, we solve for the unknowns $\frac{dF^h}{dt}$, $\frac{dF^w}{dt}$, and $\frac{dF^s}{dt}$ using the normal equations (see Appendix A for details). We then





numerically integrate $\frac{dF^h}{dt}$, $\frac{dF^w}{dt}$, and $\frac{dF^s}{dt}$ in time to obtain time series $F^w(t), F^h(t)$, and $F^s(t)$. In Sect. 3, we describe the data that we use as response functions to apply this method in practice.

### 2.2.3 Flux estimation using individual realizations

Crucially, Eq. (5) holds for the expectation of the change in observables, $\langle \Delta \mathbf{Y} \rangle(t)$. For applications to climate, this implies that it holds for the ensemble average, rather than individual realizations such as observations. Thus, for individual realizations,

we create an artificial ensemble by detrending the data, applying block bootstrapping, and adding the trend back following the methodology from McKinnon et al. (2017). For the block bootstrapping, we use a block size of 2 years as this is larger than the autocorrelation time scale of the observables. We test dependence of results on block size in Sect. 4.

We first apply linear response theory to an individual ensemble member (a single realization) as

$$\Delta \mathbf{Y_i}(t) = \int_0^t \mathbf{R}^h(t-\tau)\frac{dF_i^h}{d\tau}d\tau + \int_0^t \mathbf{R}^w(t-\tau)\frac{dF_i^w}{d\tau}d\tau + \int_0^t \mathbf{R}^s(t-\tau)\frac{dF_i^s}{d\tau}d\tau + \boldsymbol{\eta}(t), \tag{7}$$

where $\Delta \mathbf{Y_i}(t)$ is the change in the observable vector for the $i$th individual ensemble member. Here, as the observables are

not ensemble averaged, the equation includes an unknown noise term, $\boldsymbol{\eta}(t)$. The noise term has $2n$ rows, where each row corresponds to an observable (salinity or temperature in one of the $n$ mixtures). (e.g., Torres Mendonça et al., 2021). We discretize Eq. (7) in the same way as Eq. (5), and

$$\Delta \mathbf{Y}_i(t) = \sum_{k=0}^m \mathbf{R}^h(m-k)\frac{dF_i^h}{dt}(k) + \sum_{k=0}^m \mathbf{R}^w(m-k)\frac{dF_i^w}{dt}(k) + \sum_{k=0}^m \mathbf{R}^s(m-k)\frac{dF_i^s}{dt}(k) + \boldsymbol{\eta}(t), \tag{8}$$

follows. The averaged version of Eq. (7) is

$$\langle \Delta \mathbf{Y}(t) \rangle = \left\langle \int_0^t \mathbf{R}^h(t-\tau)\frac{dF_i^h}{d\tau}d\tau + \int_0^t \mathbf{R}^w(t-\tau)\frac{dF_i^w}{d\tau}d\tau + \int_0^t \mathbf{R}^s(t-\tau)\frac{dF_i^s}{d\tau}d\tau + \boldsymbol{\eta}(t) \right\rangle$$

$$= \left\langle \int_0^t \mathbf{R}^h(t-\tau)\frac{dF_i^h}{d\tau}d\tau \right\rangle + \left\langle \int_0^t \mathbf{R}^w(t-\tau)\frac{dF_i^w}{d\tau}d\tau \right\rangle + \left\langle \int_0^t \mathbf{R}^s(t-\tau)\frac{dF_i^s}{d\tau}d\tau \right\rangle \tag{9}$$

where $\langle . \rangle$ is the ensemble average over all ensemble members and $\langle \boldsymbol{\eta}(t) \rangle = 0$. We assume that Eq. (5) and Eq. (9) are approxi-

mately equivalent; in other words, we assume that $[\langle \frac{dF_i^h}{dt} \rangle, \langle \frac{dF_i^w}{dt} \rangle, \langle \frac{dF_i^s}{dt} \rangle] \approx [\frac{dF^h}{dt}, \frac{dF^w}{dt}, \frac{dF^s}{dt}]$.

For each member of an artificial ensemble created by block bootstrapping, we solve Eq. (8) for $[\frac{dF_i^h}{dt}, \frac{dF_i^w}{dt}, \frac{dF_i^s}{dt}]$, ignoring the effect of $\boldsymbol{\eta}(t)$. We estimate the mean $[\frac{dF^h}{dt}, \frac{dF^w}{dt}, \frac{dF^s}{dt}]$ with associated error by taking the average and standard deviation across artificial ensemble members of $[\frac{dF_i^h}{dt}, \frac{dF_i^w}{dt}, \frac{dF_i^s}{dt}]$ as Eq. (9) implies.

Throughout the next two sections (Sects. 3 and 4), we introduce data products for application of the method in practice, for

validation, and for rescaling of our freshwater flux change estimate in terms of a percentage change of the hydrological cycle per degree C. These data will be explained in text in detail as they are introduced; however, for convenience a summary is in Table 1.



**Table 1.** A summary of data products used throughout the paper for: application of the method, validation, and rescaling our final result as a percentage change of the hydrological cycle per degree C. The data listed here are described in more detail in Sects. 3 and 4.

| Use | Data product |
| --- | --- |
| Response functions ($\mathbf{R}^h(t)$, $\mathbf{R}^w(t)$, $\mathbf{R}^s(t)$ in Sect. 2) | Ocean only Flux-Anomaly-Forced Model Intercomparison Project (FAFMIP) (Gregory et al., 2016; Todd et al., 2020) |
| Applying method to model temperature and salinity data and comparing against known freshwater fluxes (Sect. 3) | Community Earth System Model (CESM) Large Ensemble (Kay et al., 2015) |
| Estimating hydrological cycle change from observations over 1975 to 2019 (Sect. 4) | Ocean surface temperature and salinity from the Institute of Atmospheric Physics (IAP) (Cheng and Zhu, 2016; Cheng et al., 2020) |
| Express observational estimate as a percentage change of the climatological hydrological cycle strength per degree of warming (Sect. 4) | Estimating the Circulation and Climate of the Ocean (ECCO) (Forget et al., 2015; ECCO Consortium et al., 2021; ECCO Consortium et al.) and surface air temperature from NASA GISS (Lenssen et al., 2019; GISTEMP Team, 2023) |

## 3 Validation of surface flux estimates from linear response theory in CESM

In this section, we use 34 members of the CESM large ensemble (ensemble members 001-035) over the period 1975 to 2019 to develop, test, and validate the methodology of surface flux estimation. The forcing used in the simulation is historical until 2005 and then RCP8.5 onwards (Kay et al., 2015). The CESM ensemble mean freshwater flux change over the period considered is shown in Fig. 2a.

### 3.1 Extracting observables and response functions

To select the main regions defining the salinity pattern (and our observables), we fit a GMM to the surface salinity distribution of the CESM ensemble mean over 1975 to 1980 (see Sect. 2.2.1). We include all points south of 65N; the Arctic is excluded for simplicity due to its small size and to remove the potential effect of sea ice melt on salinity. We choose the number of mixtures as 6, as this is the smallest number where the gradient of AIC and BIC relative to the number of mixtures flattens (i.e., the second derivative approaches 0) as shown in Fig. 1b. The GMM fit to the CESM ensemble mean data is shown in Fig. 3a and b.

To estimate the response functions, we introduce data from the ocean only Flux-Anomaly-Forced Model Intercomparison Project (FAFMIP). FAFMIP will be used to extract $\mathbf{R}^h(t)$, $\mathbf{R}^w(t)$, and $\mathbf{R}^s(t)$ as defined in Sect. 2. In the ocean FAFMIP experiments, heat flux, freshwater flux, and wind stress perturbations associated with $CO_2$ doubling are imposed separately on ocean models (Gregory et al., 2016; Todd et al., 2020). The FAFMIP freshwater flux perturbation, for example, imposed as a step function is shown in Fig. 2b. Data from FAFMIP is well suited to our method because forcings are imposed both as





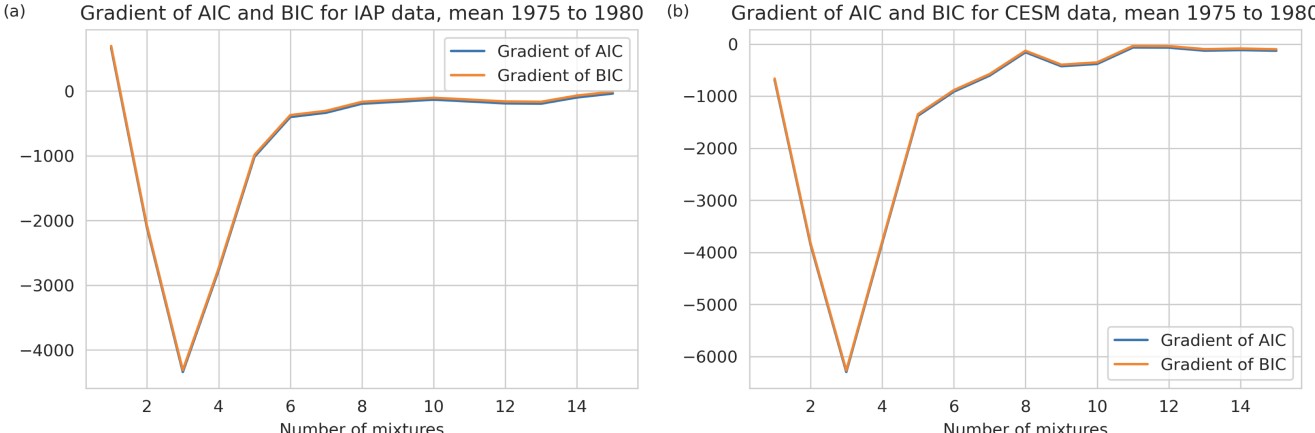

**Figure 1.** The AIC and BIC derivatives relative to the number of mixtures plotted against the number of mixtures. Panel a uses the IAP surface salinity distribution south of 65N over the period 1975 to 1980, while panel b uses the CESM ensemble mean surface salinity distribution over the same region and time period. The plots of AIC and BIC versus number of mixtures are in the supplement (Fig. S1).

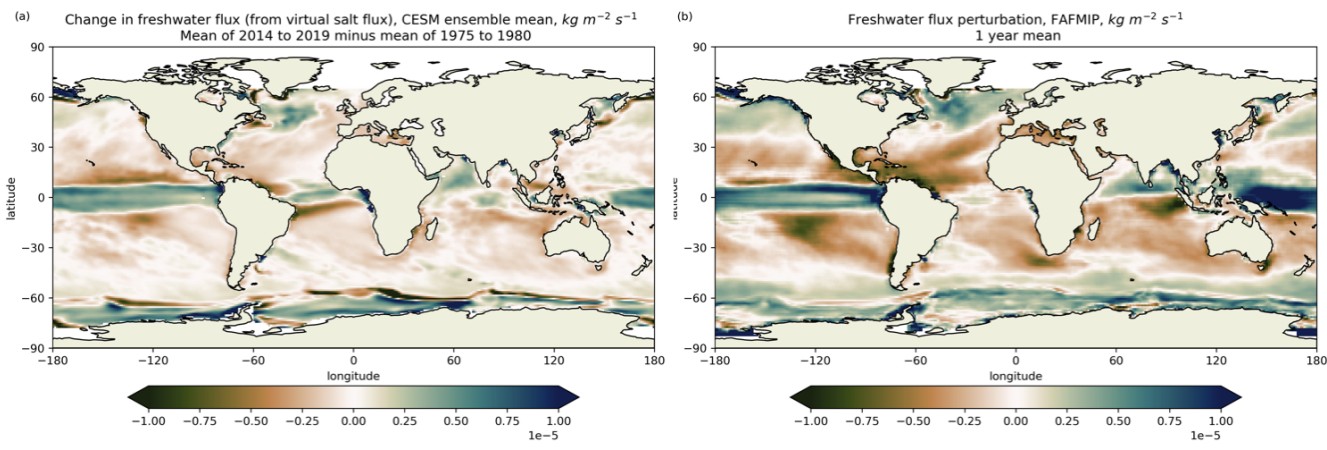

**Figure 2.** Change in ensemble mean CESM freshwater fluxes over 1975 to 2019 (panel a) and the annual mean freshwater flux perturbation map from FAFMIP (panel b) (Gregory et al., 2016; Todd et al., 2020). The FAFMIP map is $\mathbf{N}(x, y)$ in Eq. (10).



**Figure 3.** GMMs fit to the mean surface salinity distribution (a and c) and location of mixtures based on these models (b and d). The top row (a and b) is using surface salinity data from the CESM large ensemble with a mean across 34 members of the ensemble and over the period 1975 to 1980 (Kay et al., 2015). The bottom row (c and d) is using surface salinity data from the IAP (Cheng et al., 2020) with a mean over the period 1975 to 1980.





step functions and as individual components (see Eq. (4) and (5)). Thus, the salinity and temperature responses to FAFMIP perturbations are used in our method as the response functions ($\mathbf{R}^h(t)$, $\mathbf{R}^w(t)$, and $\mathbf{R}^s(t)$) in Eq. (5). There is a 15-year relatively fast response of the ocean state after applying the FAFMIP abrupt forcing, before a slow and steady linear response to the forcing which we are trying to capture. Therefore, we truncate the time series of the response functions such that $\mathbf{R}^h(0)$, $\mathbf{R}^w(0)$, and $\mathbf{R}^s(0)$ are taken to be between 15 and 20 years after the forcing is applied. Thus, in practice, the linear response theory expressions (Eq. (6) and (8)) are solved for each adjustment time (between 15 and 20 years) and the mean is taken.

### 3.2   Impact of ocean transport and linearity of forcings on surface salinity

At the end of Sect. 1, we hypothesized that global heat fluxes affect regional surface salinity differently. Here, we validate this hypothesis by examining the effect of the FAFMIP step forcings – heat flux, freshwater flux, and wind stress change – on surface salinity in GMM regions found in Fig. 3a and b. Figure 4 shows the change in each mixture region between the last

decade of a forcing experiment and the last decade of control; panels a-c show results for different FAFMIP ocean models. We find that the freshwater flux forcing makes fresher (saltier) regions fresher (saltier) with a near linear scaling of change relative to the region's climatological salinity. The heat flux perturbation tends to make saltier regions (mixtures 5 and 6) saltier with little effect in other regions. The wind stress change has little effect on the salinity pattern. Figure S4 (in the supplement) shows comparable plots for surface temperature, but we focus on salinity here as it is the primary tracer affected by freshwater fluxes. We confirm that circulation changes, induced by heat fluxes, are regional and imprint on surface salinity with a different spatial pattern than freshwater fluxes. Thus, our methodology, which uses regional tracer changes can account for the distinct effects of heat fluxes and freshwater fluxes when inferring surface freshwater flux changes.

We invoke two different linearity assumptions in our method. The first assumption of linearity in the methodology is that regional ocean salinity and temperature changes under anthropogenic forcing can be separated into the sum of the effect

from freshwater fluxes, heat fluxes, and wind stress change imposed separately. As above, we evaluate this using the effect of FAFMIP step forcings on ocean regions defined by a GMM fit to the CESM ensemble mean salinity distribution (mixture locations in Fig. 3b). Figure 5 compares the change in salinity (panel a) and temperature (panel b) due to separate imposition of forcings ('faf-stress+faf-water+faf-heat') and due to forcings applied at once ('faf-all'). We examine agreement by finding the $L^2$ norm of the response in the 6 regions due to forcings imposed individually divided by the $L^2$ norm of the response from the combined forcing. These values for surface salinity are: 0.916 for MITgcm, 0.959 for ACCESS-OM2, and 1.340 for HadOM3. For surface temperature, these values are: 0.974, 0.962, and 1.118. We see good agreement between the sum of individual forcings and forcings applied together. A notable exception is the sixth mixture for the HadOM3 model where there is a non-linear response. Overall, the agreement is sufficiently close to use the linearity approximation, but the disagreement in the sixth mixture for HadOM3 is a caveat of this work.

The second assumption of linearity in the method is that linear response theory holds – i.e. that the change in observables is linear with respect to the forcing strength. In particular, we assume that this holds as in Eq. (5) where linear response theory is applied to individual forcing components. The available data where individual forcing components are imposed on ocean models comes from FAFMIP; however, the FAFMIP experiments impose perturbations only as step forcings so we cannot

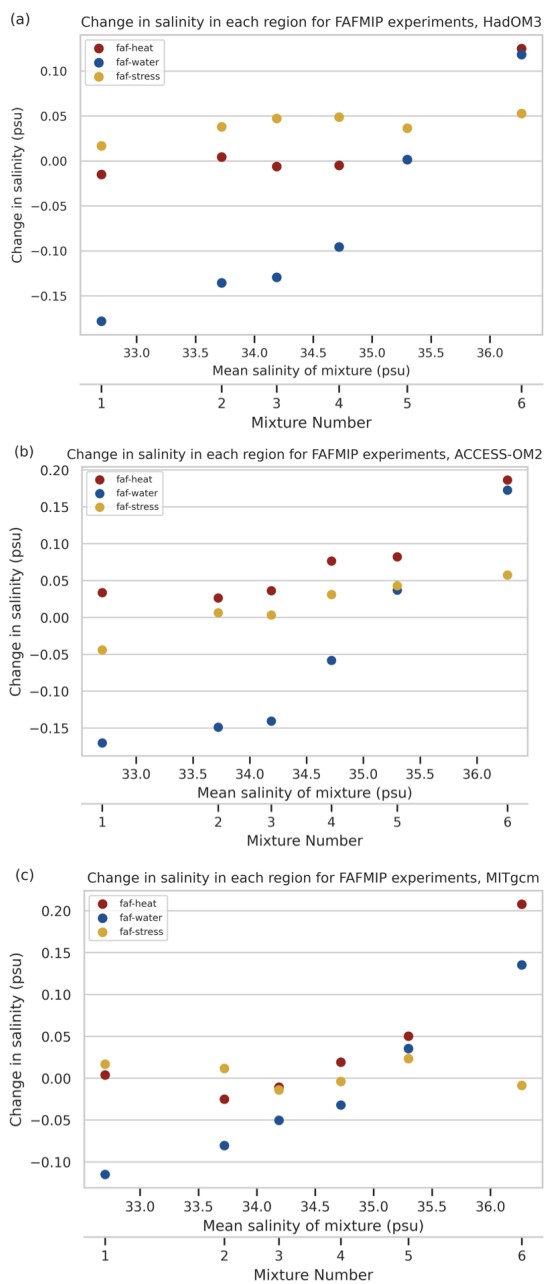

**Figure 4.** The change in surface salinity in each region from the GMM applied to the CESM ensemble mean data (Fig. 3b) for each individual forcing experiment. The heat flux experiment (faf-heat) is shown in red, freshwater flux (faf-water) in blue, and wind stress perturbation (faf-stress) in yellow. The response is defined as the difference between the last decade of a forced run and the last decade of the control run. Panel a-c show the results for ocean models HadOM3, ACCESS-OM2, and MITgcm respectively.

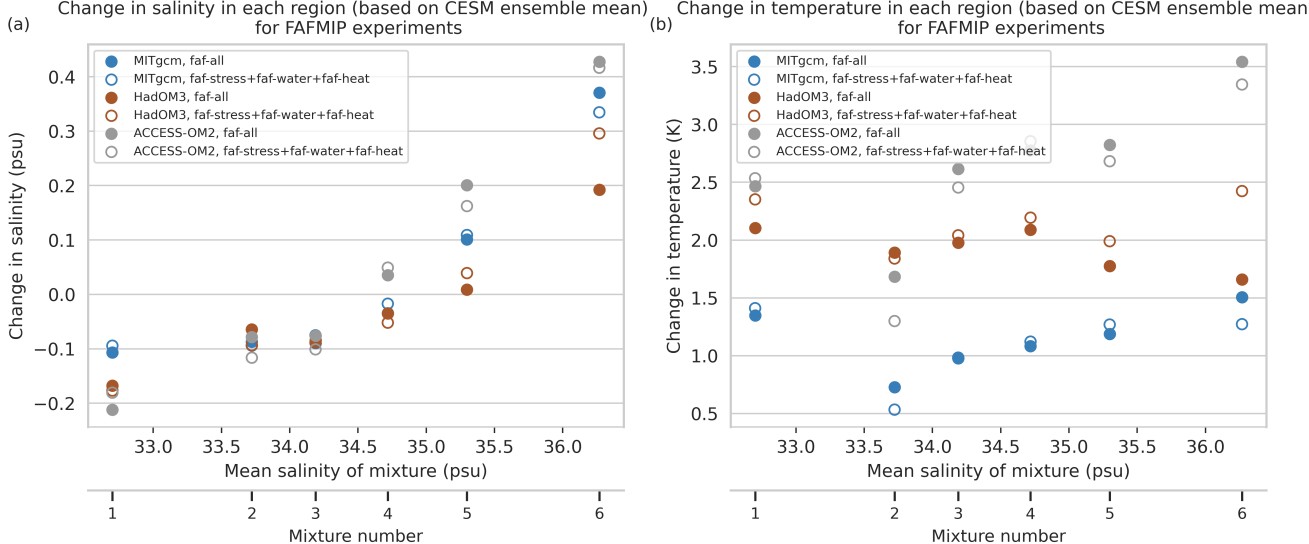

**Figure 5.** Comparing the surface salinity response (a) and surface temperature response (b) in each region between two cases: the sum of the response when forcings are applied individually (faf-stress+faf-water+faf-heat) and the response to all forcings applied at once (faf-all). The response is defined as the difference between the last decade of a forced run and the last decade of the control run. The response is largely linear with the exception of the sixth mixture for the HadOM3 model.

directly evaluate linearity with respect to forcing strength. Instead, in the next subsections, we validate the full methodology
by testing if the true freshwater fluxes from model data can be recovered via our method using only surface salinity and temperature time series.

### 3.3   Estimating surface flux amplification from CESM ensemble mean data

We test the method on the CESM ensemble mean over the period 1975 to 2019. We apply Eq. (6) where each row of $\langle \Delta \mathbf{Y} \rangle(t)$ is a time series of (normalized and area weighted) salinity and temperature in each GMM region from the ensemble mean data.
The non-normalized or area weighted versions of the observables are plotted in Fig. 6. We solve Eq. (6) for the unknown time series of each forcing as a proportion of the step function strength (see Fig. 7 for $F^w(t)$). We take the change over the period to be the mean over the last 5 years, 2015 to 2019, minus the mean over the first 5 years, 1975 to 1979. We find that the change in freshwater fluxes is $0.3240$ times the FAFMIP step forcing.

For comparison, we quantify the true response of E-P pattern change by finding the change in the model's freshwater fluxes
over both precipitation and evaporation dominated regions. We separately integrate the annual mean freshwater fluxes over the region where the FAFMIP perturbation is positive (precipitation dominated) or negative (evaporation dominated) and then scale by the magnitude of the FAFMIP pattern calculated in the same way. We define the precipitation or evaporation dominated regions using the FAFMIP perturbation because the linear response theory method finds a scaling of this pattern. The target

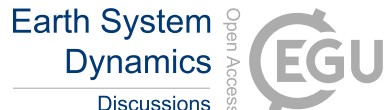

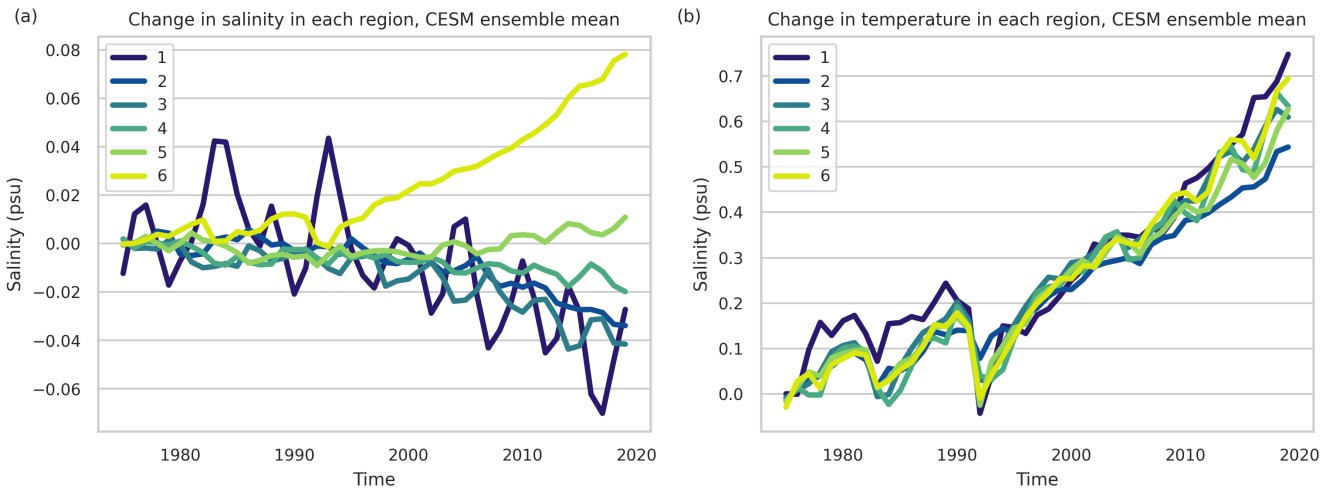

**Figure 6.** The change in surface salinity (a) and surface temperature (b) in each region from CESM ensemble data over the period 1975 to 2019. Here, the data is an ensemble mean over 34 members and the anomaly is taken from the mean of the first two years. The locations of the mixtures are shown in Fig. 3b.

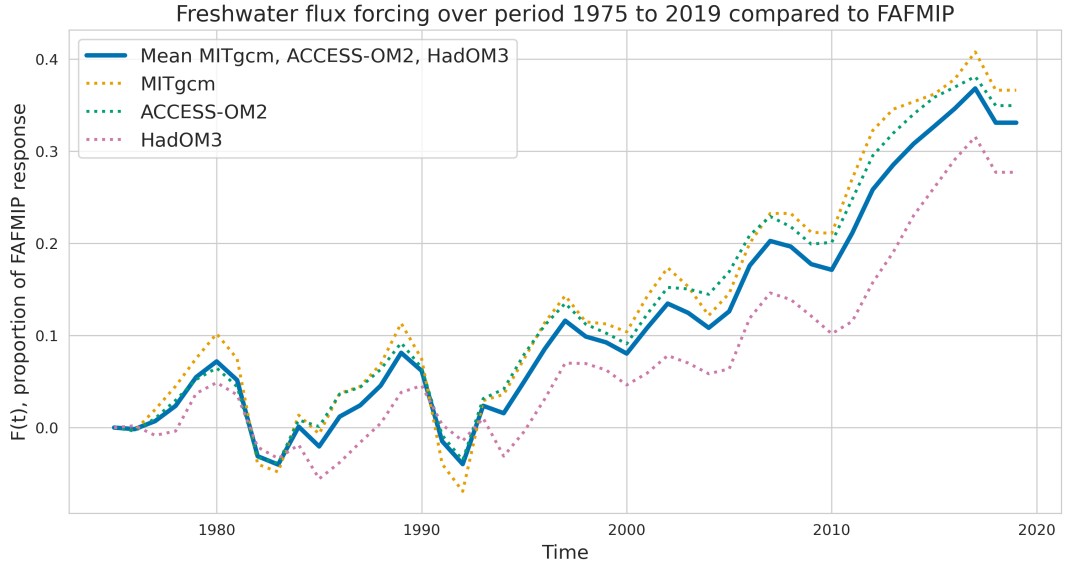

**Figure 7.** $F^w(t)$ found by solving Eq. (6), where $\langle \Delta \mathbf{Y} \rangle(t)$ is regional salinity and temperature from the CESM ensemble mean over the period 1975 to 2019.



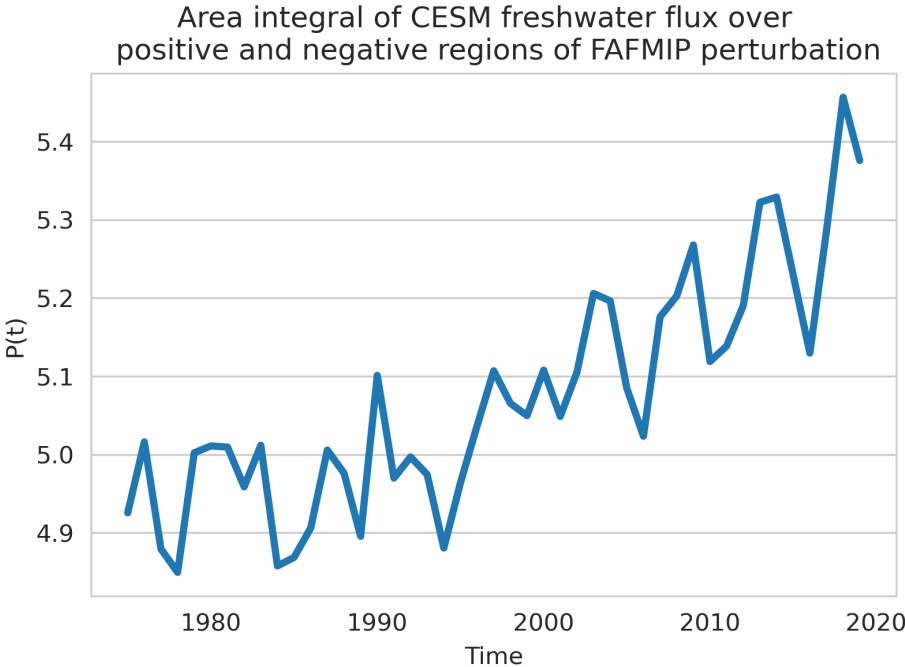

**Figure 8.** $P(t)$, the target truth metric, as defined in Eq. (10) over the period 1975 to 2019 for CESM ensemble mean data.

metric is written as

$$P(t) = \frac{\int_{\Omega_-} \mathbf{C}(x,y,t)dA + \int_{\Omega_+} \mathbf{C}(x,y,t)dA}{\int_{\Omega_-} \mathbf{N}(x,y)dA + \int_{\Omega_+} \mathbf{N}(x,y)dA} \tag{10}$$

where $\mathbf{C}(x,y,t)$ is the freshwater flux pattern from CESM at time $t$, $\mathbf{N}(x,y)$ is the FAFMIP freshwater perturbation pattern (see Fig. 2b), and $\Omega_-$ and $\Omega_+$ are regions where $\mathbf{N}(x,y)$ is negative and positive respectively.

We account for internal variability in $P(t)$ using block bootstrapping on the time series (Fig. 8) with 2 year blocks and 3000 members (McKinnon et al., 2017). We find the mean and standard deviation of the change in the freshwater fluxes using $P(t)$, where the change is quantified as the mean over the last 5 years minus the mean over the first 5 years. We find that the true change in freshwater fluxes for the CESM ensemble mean is $0.3733 \pm 0.0552$ times the FAFMIP perturbation. Thus, the result of $0.3240$ from our method is within error bounds of the truth.

### 3.4 Estimating surface flux amplification from individual ensemble members of CESM

We test the application of the method on individual CESM ensemble members. For this application, the dimensionality reduction step is done for each member by fitting a GMM to the salinity distribution of the individual realization; this results in region categorization analogous to Fig. 3b for each individual member. We create an artificial ensemble around each member. For the $i^{\text{th}}$ (artificial) member, we solve Eq. (8) for $[\frac{dF_i^h}{dt}, \frac{dF_i^w}{dt}, \frac{dF_i^s}{dt}]$. We then take the mean and standard deviation of the

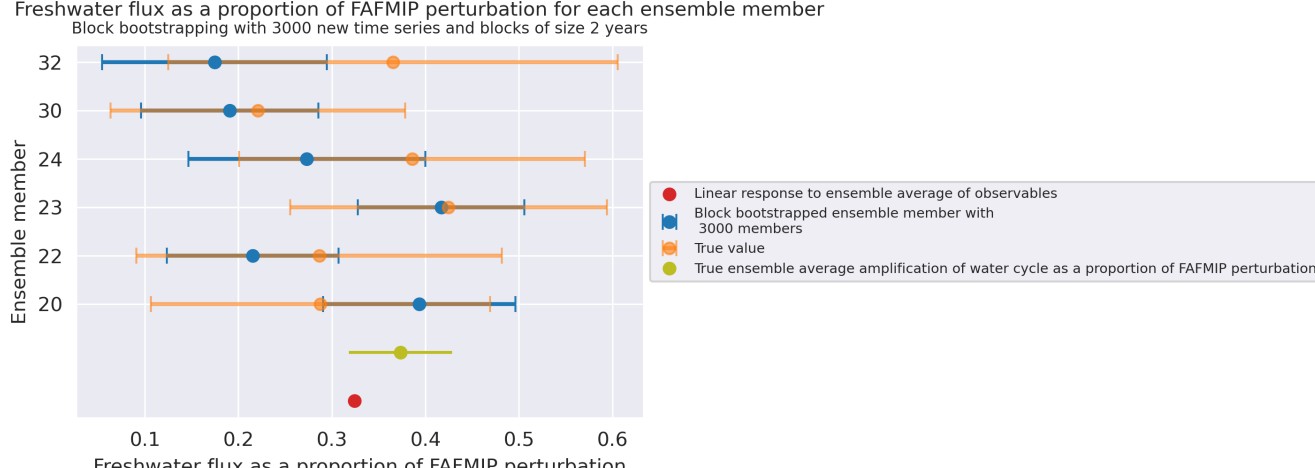

**Figure 9.** Freshwater flux responses for CESM ensemble members following methodology described in text (blue) compared to truth from E-P model fields (orange). Here, we plot only members which met salinity trend significance criteria defined in text. For members meeting the significance criteria, the true amplification of freshwater fluxes is captured within error bars. The tests on all ensemble members are available in Fig. S5 in the supplement.

change in $F_i^w(t)$ (last 5 years minus first 5 years) across all artificial ensemble members to find freshwater flux change with error (see Sect. 2.2.3).

Applying this methodology, we find the method does not capture the true response for all individual CESM members (see Fig. S5 in the supplementary). However, while the CESM members have significant temperature trends over the historical period, many members have insignificant (linear) salinity trends due to variability. In fact, none of the members have all regional linear trends significant at the $p < 0.05$ level. We impose significance criteria by requiring regions with strong trends across ensemble members, regions 2 and 6, have $p < 0.05$. We require that other regions' trends have $p < 0.18$ – i.e. insignificant

but the internal variability to signal ratio is capped. We exclude restrictions on region 4, as for most members it has a large amount of internal variability but doesn't affect the result, likely because of its small size. We find 6 members which meet these statistical significance criteria. For these members, the freshwater flux amplification found by the linear response theory method is plotted in Fig. 9 and compared against the true amplification, determined by the change in Eq. (10).

Thus, we find that the method, as outlined in Sect. 2, can capture the true freshwater amplification for individual mem-

bers provided the salinity trends meet certain significance criteria. However, our estimate captures less of the amplification uncertainty than the target metric; thus, we propose interpreting the error bars from our method as a lower bound on error. It is sensible that we need significance criteria on salinity trends, as members not meeting them have observables (salinity) dominated by internal variability rather than forced response. This same process is tested in the supplement over the period 2011 to 2055 where salinity trends are stronger and thus most members reach the significance criteria. In that case, we recover

the true result for $90.3\%$ of members.





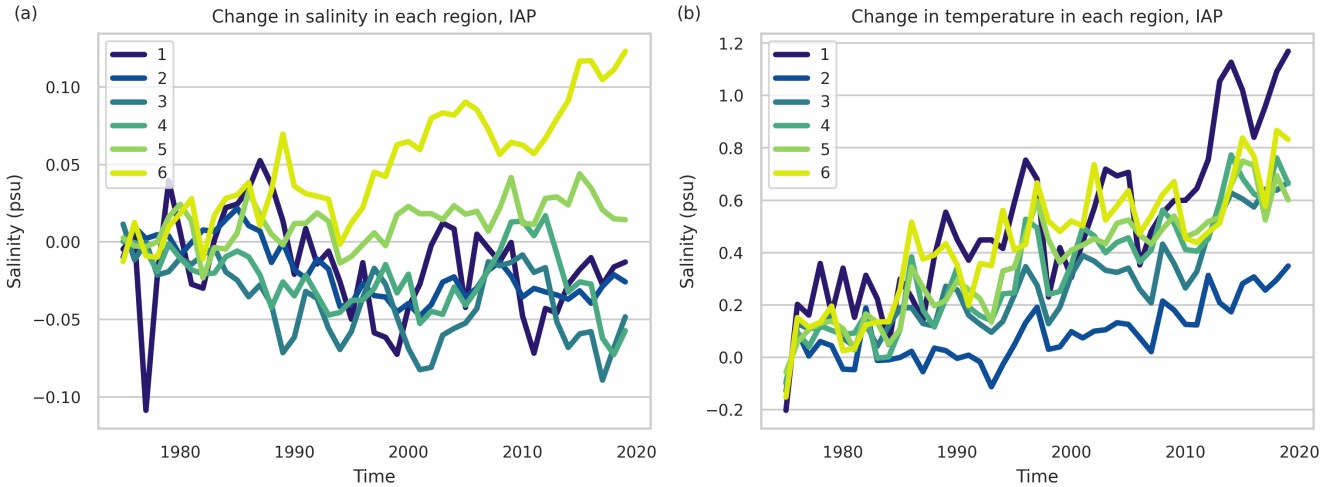

**Figure 10.** The change in surface salinity (a) and surface temperature (b) in each region from the IAP data over the period 1975 to 2019. Here, the change is taken from the mean of the first two years. The locations of the mixtures are shown in Fig. 3d.

## 4    Estimating Freshwater Fluxes from Observations

In this section, we apply our methodology (outlined in Sects. 2 and 3) to the Institute of Atmospheric Physics (IAP) dataset over the period 1975 to 2019 (Cheng and Zhu, 2016; Cheng et al., 2020). We focus on the IAP data as other available observational datasets are either limited in time or have salinity trend biases. In particular, the dataset following Ishii et al. (2005) ends

in 2012, while EN4 data (Good et al., 2013) has salinity biases compared to other data; these biases, for example, lead to freshwater flux estimates different than when using other data products in Sohail et al. (2022) (see their Fig. 3).

Based on AIC and BIC, we use 6 mixtures for the GMM (see Fig. 1a) when performing the dimensionality reduction step of the method. The change in temperature and salinity in each region as found by the GMM (see Fig. 3c and d) are shown in Fig. 10. We then create an artificial ensemble around the original data as required to apply the methodology to a single realization.

We find that the observations meet the salinity trend significance criteria outlined in Sect. 3 by a large margin. The p-values of linear salinity trends are $6.33 \times 10^{-2}$, $7.55 \times 10^{-9}$, $1.90 \times 10^{-3}$, $6.34 \times 10^{-2}$, $7.75 \times 10^{-6}$, and $3.47 \times 10^{-27}$ for each region respectively. Thus, the significance criteria based on testing on the CESM large ensemble are met, and the full methodology should recover the true freshwater flux amplification.

From 1975 to 2019, we find the freshwater flux response was $0.3000 \pm 0.0803$ times the FAFMIP perturbation. This value

is insensitive to choices made in the methodology. Table 2 shows the resultant estimate if the methodology had instead been carried through with 5 or 7 mixtures. We also evaluate the sensitivity of the estimate to choices in the block bootstrapping step and find it is insensitive to block size (Fig. C1) and number of artificial ensemble members created (Fig. C2). Thus, we find that the method estimates freshwater fluxes from observational data robustly.





**Table 2.** The response as a proportion of the FAFMIP freshwater flux perturbation using different numbers of mixtures in the GMM. Here the method is applied to the IAP data over the period 1975-2019.

| Number of mixtures | Response as a proportion of FAFMIP perturbation |
|---|---|
| 5 | $0.3103 \pm 0.0769$ |
| 6 | $0.3000 \pm 0.0803$ |
| 7 | $0.2915 \pm 0.0799$ |

We also note that the hypotheses and assumptions motivating and utilized in the method as tested in Sect. 3 similarly hold for the IAP data. Figures 4 and 5 which used the GMM regions from CESM have equivalents using GMM regions from the IAP in the supplement (Fig. S2 and S3).

Our estimate is expressed as a scaling of the FAFMIP perturbation. To facilitate comparison with previous studies, we now convert it to a percentage change of the climatological hydrological cycle. This conversion introduces more error into the estimate as it requires scaling the result by other datasets. Here, we use mean freshwater fluxes from the Estimating the Circulation and Climate of the Ocean (ECCO) state estimate as the climatological hydrological cycle strength (Forget et al., 2015; ECCO Consortium et al., 2021; ECCO Consortium et al.) (see Appendix B). We find that our estimate as a proportion of the FAFMIP perturbation is equivalent to hydrological cycle amplification of $4.027 \pm 1.078\%$. We can also express this as a percentage change per degree Celsius change in surface air temperature. We estimate the change in surface air temperature over the period of interest from NASA GISS data (Lenssen et al., 2019; GISTEMP Team, 2023) by taking the difference between the means over 2017 to 2021 (2 years before and after 2019) and over 1973 to 1977. Scaling by the surface temperature change, we find an amplification of $4.515 \pm 1.209\%$ $^{o}C^{-1}$ (Fig. 11).

Although we focus on freshwater fluxes, our methodology also finds scalings of the other FAFMIP forcings. From the IAP data, we find a heat flux change of $0.3993 \pm 0.1277$ times the FAFMIP perturbation. Integrating the FAFMIP heat flux forcing south of 65N, we find a heat flux increase over 1975 to 2019 of $0.5578 \pm 0.1784$ W m$^{-2}$. This is within error bounds of other estimates. The AR6 reports a heat flux increase of $0.52 \pm 0.15$ W m$^{-2}$ over 1971 to 2018 (Gulev et al., 2021), while Cheng et al. (2022) found $0.44 \pm 0.08$ W m$^{-2}$ over the same period.

## 5 Conclusions

Accurate estimates of the amplification of the hydrological cycle over the historical period is important for future projections. However, freshwater flux trends cannot be measured directly and trends must be inferred, generally from ocean salinity changes. Here, we introduced a method, based on linear response theory, that infers freshwater fluxes from the evolving surface salinity pattern, building on previous studies (Durack et al., 2012; Zika et al., 2018). Our methodology flexibly accounts for regional

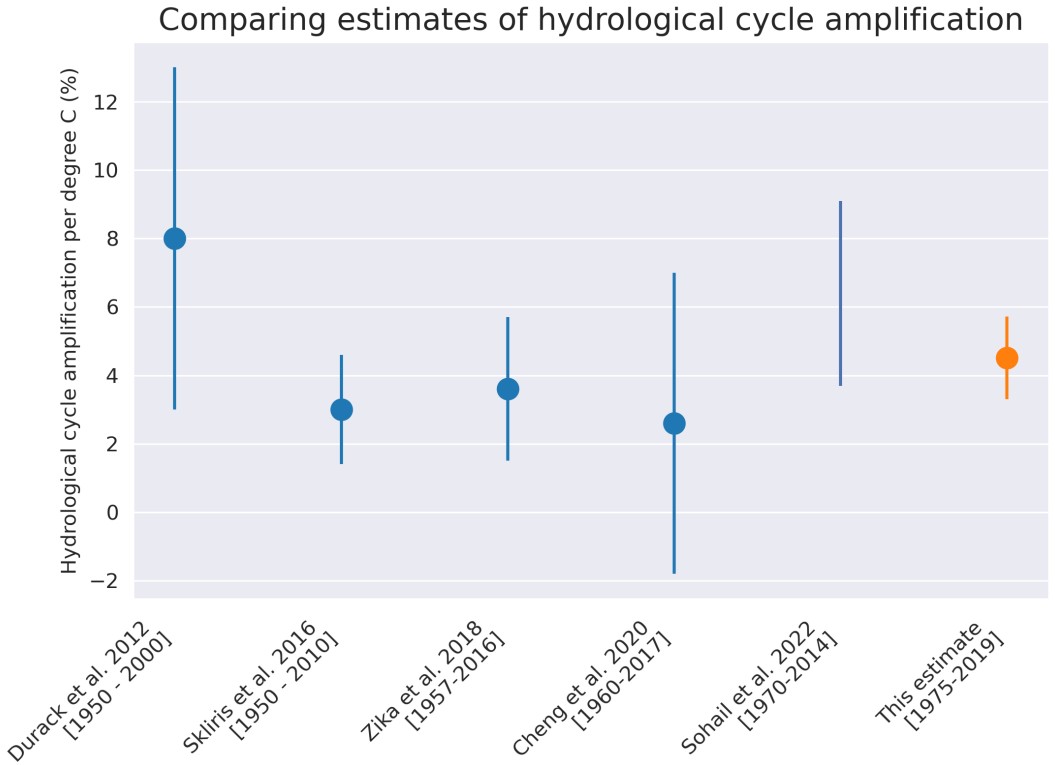

**Figure 11.** Comparing this estimate of hydrological cycle amplification with previous studies. Here, we scaled the Sohail et al. (2022) estimate using the change in surface air temperature over the period 1970 to 2014 from NASA GISS data so it could be compared against the other estimates shown (GISTEMP Team, 2023). We take the change in surface air temperature as the mean over 2012 to 2016 minus the mean over 1968 to 1972.

changes in surface salinity due to different forcings (heat flux, freshwater flux, wind stress), therefore relaxing previously used assumptions imposing linear scaling of the climatological salinity pattern.

We validated our methodology using data from the CESM large ensemble. We recovered the true freshwater flux change

(as determined from the model E-P) for CESM ensemble average data over 1975 to 2019. We similarly recovered the true freshwater flux change for individual ensemble members, provided significance criteria on salinity trends were met. Thus, our method performs well on test data and can be applied to observations.

The recovery of the true freshwater flux change for individual ensemble members implies that linear response theory, under certain circumstances, can be applied to individual realizations (e.g., observations) by creating an artificial ensemble following

McKinnon et al. (2017). This idea could be used for other applications of linear response theory in climate science, which has previously been limited by the need for an ensemble average, and adds to previous literature on linear response theory applied to climate (Lucarini and Sarno, 2011; Lembo et al., 2020; Ragone et al., 2016; Torres Mendonça et al., 2021).



Over the period 1975 to 2019, from ocean tracer observations, we estimate a change in surface freshwater fluxes equivalent to $0.3000 \pm 0.0803$ times the FAFMIP perturbation. For comparison with previous studies, we convert this to $4.515 \pm$

$1.209\%$ $^{o}\mathrm{C}^{-1}$, scaling by the ECCO state estimate as the climatological hydrological cycle. After this scaling, our result is largely in agreement with previous work (Skliris et al., 2016; Zika et al., 2018; Cheng et al., 2020; Sohail et al., 2022). We also estimate a heat flux increase of $0.5578 \pm 0.1784$ W m$^{-2}$ in agreement with previous studies (Gulev et al., 2021; Cheng et al., 2022). In this work, we accounted for the regional imprint of ocean circulation on salinity, yet found a similar value of hydrological cycle amplification as Zika et al. (2018) which assumed the impact of heat fluxes on the surface salinity pattern is a

linear scaling of the climatological salinity pattern. This implies that over the time period considered, it may be sufficient to use this assumption of scaling the existing pattern. However, as ocean circulation changes increase due to increased anthropogenic forcing, this assumption may not hold, and our method which accounts for regional changes can be employed.

Our estimate of hydrological cycle amplification is subject to a number of caveats indicating the methodology may underestimate error. The error bars reported should be interpreted as a lower bound on error due to the following sources: 1) we do

not quantify the uncertainty associated with utilizing different ocean models as the response functions; 2) we do not account for error in the salinity and temperature observations themselves; 3) the artificial ensemble is not equivalent to a true ensemble average; 4) the method assumes that the change in freshwater fluxes in observations is well approximated by scaling the magnitude of the FAFMIP perturbation. Additionally, our assumption that the impact of total anthropogenic forcing can be broken into the summation of the impact of heat fluxes, freshwater fluxes, and wind stress change holds, except for the response in the

saltiest region in the HadOM3 model; this exception may slightly bias results. Despite these caveats, our method performed well on CESM test data, and our estimate based on observations agrees with previous studies. Thus, this work, which accounts for regional effects of ocean transport change, adds confidence to the conclusion that the hydrological cycle is amplifying at a rate less than Clausius-Clapeyron. This result warrants further investigation into the mechanisms controlling the hydrological cycle amplification rate and the representation in models.

*Code and data availability.*  The code used for this work is available at https://github.com/aurora-bf/freshwater_flux_linresp (Basinski-Ferris, 2023). The data from the ocean-only FAFMIP project is available at https://gws-access.ceda.ac.uk/public/ukfafmip. The data that the methodology is applied to is available publicly. The CESM large ensemble data is available at https://www.earthsystemgrid.org/dataset/ucar.cgd.ccsm4.cesmLE.ocn.proc.monthly_ave.html, the IAP data is available at http://www.ocean.iap.ac.cn/pages/dataService/dataService.html, and the NASA surface air temperature data is available at https://data.giss.nasa.gov/gistemp/.





## Appendix A: More details on discretization and solution of linear response problem

In the case of an ensemble average of observables, the equation that we solve at each time step to find $\left[\frac{dF^h}{dt}, \frac{dF^w}{dt}, \frac{dF^s}{dt}\right]$ is given in the main text (Eq. (6)). Here, we show the explicit steps for solution. First, we rewrite the equation as

$$\langle \Delta \mathbf{Y}\rangle(t) - \sum_{k=0}^{m-1}\mathbf{R}^h(m-k)\frac{dF^h}{dt}(k) - \sum_{k=0}^{m-1}\mathbf{R}^w(m-k)\frac{dF^w}{dt}(k) - \sum_{k=0}^{m-1}\mathbf{R}^s(m-k)\frac{dF^s}{dt}(k)$$
$$= \mathbf{R}^h(0)\frac{dF^h}{dt}(m) + \mathbf{R}^w(0)\frac{dF^w}{dt}(m) + \mathbf{R}^s(0)\frac{dF^s}{dt}(m),$$

so that at time step $m$, the left side has only known information and unknowns are on the right side. This could be expressed as

$$\langle \Delta \mathbf{Y}\rangle(t) - \sum_{k=0}^{m-1}\mathbf{R}^h(m-k)\frac{dF^h}{dt}(k) - \sum_{k=0}^{m-1}\mathbf{R}^w(m-k)\frac{dF^w}{dt}(k) - \sum_{k=0}^{m-1}\mathbf{R}^s(m-k)\frac{dF^s}{dt}(k) = \mathbf{A}*x, \tag{A1}$$

where $\mathbf{A}$ is a $2n \times 3$ matrix with each column one of the $\mathbf{R}(0)$ vectors and $x$ a 3x1 vector of $\left[\frac{dF^h}{dt}, \frac{dF^w}{dt}, \frac{dF^s}{dt}\right]^T$. This is an overdetermined system of the form $b = \mathbf{A}x$ and we solve for $x$ in the least squares sense, using the normal equations.

The method of solution is similar in the case of observations (an individual ensemble member) for which we have generated artificial ensembles. Here, for each $i$th artificial ensemble member, we rearrange Eq. (8) as

$$\Delta \mathbf{Y}_i(t) - \sum_{k=0}^{m-1}\mathbf{R}^h(m-k)\frac{dF_i^h}{dt}(k) - \sum_{k=0}^{m-1}\mathbf{R}^w(m-k)\frac{dF_i^w}{dt}(k) - \sum_{k=0}^{m-1}\mathbf{R}^s(m-k)\frac{dF_i^s}{dt}(k)$$
$$= \mathbf{R}^h(0)\frac{dF_i^h}{dt}(m) + \mathbf{R}^w(0)\frac{dF_i^w}{dt}(m) + \mathbf{R}^s(0)\frac{dF_i^s}{dt}(m) + \boldsymbol{\eta}(t),$$

so that again, at time step $m$, the left side contains known information and unknowns are on the right. We solve this equation in the same way as with Eq. (A1) in the least squares sense for $\left[\frac{dF_i^h}{dt}, \frac{dF_i^w}{dt}, \frac{dF_i^s}{dt}\right]$, except we neglect $\boldsymbol{\eta}(t)$. Then, as described in Sect. 2, we approximate the forcings $\left[\frac{dF^h}{dt}, \frac{dF^w}{dt}, \frac{dF^s}{dt}\right]$ as the ensemble average of $\left[\frac{dF_i^h}{dt}, \frac{dF_i^w}{dt}, \frac{dF_i^s}{dt}\right]$ as Eq. (9) implies.

## Appendix B: Representing freshwater flux perturbation as percentage change

The methodology described in Sects. 2 and 3 solves for the freshwater flux forcing as a proportion of the FAFMIP perturbation field. To compare with other estimates, this is converted to a percent change by comparing against a climatological value from the ECCO state estimate. We quantify the strength of the E-P pattern by taking the area integral of the absolute value of the annual mean pattern (i.e., for the FAFMIP perturbation, the integral of the absolute value of the map in Fig. 2b). This yields a value in kg s$^{-1}$ that can be converted to a value in Sverdrups (Sv). We quantify the strength of the climatological pattern by taking the mean of the ECCO freshwater fluxes from 1992 to 2001 and taking the area integral in the same way. Thus, the result can be written as a percentage change, by using

$$\frac{a \int_\Omega |\mathbf{N}(x,y)|dA}{\int_\Omega |\mathbf{ECCO}(x,y)|dA}$$

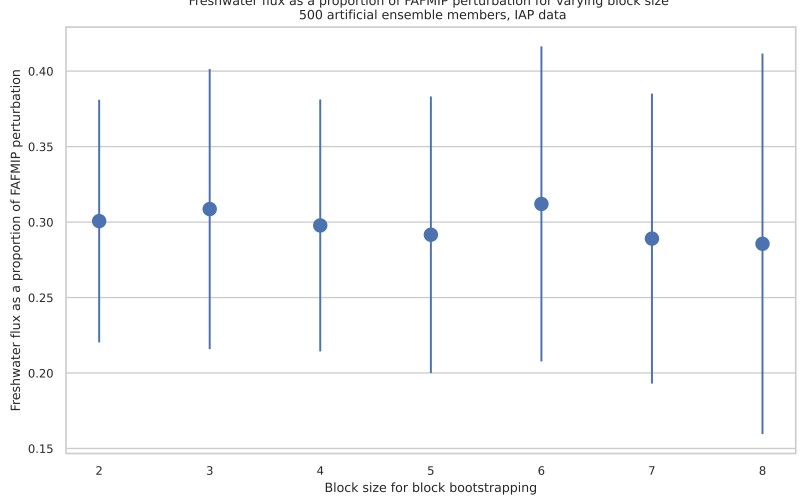

**Figure C1.** The freshwater flux change as a proportion of the FAFMIP perturbation (y axis) versus block size used when generating the artificial ensemble (x axis). Here, we applied the method to the IAP data over the period 1975 to 2019 and used artificial ensembles with 500 members.

where $\Omega$ is the ocean surface south of 65N, $\mathbf{N}(x,y)$ is the map of the annual mean FAFMIP freshwater fluxes perturbation, $\mathbf{ECCO}(x,y)$ is the map of the ECCO freshwater fluxes averaged over the period 1992 to 2001, and $a$ is the proportion of the FAFMIP flux field as found by linear response theory. We can express this quantity per degree C by dividing by the change in surface air temperature over the period 1975 to 2019 (GISTEMP Team, 2023).

**Appendix C: Sensitivity to block bootstrapping choices**

Here, we show results for the sensitivity of the estimate from observations (see Sect. 4) to choices in the block bootstrapping step of the methodology. As discussed in text, we find the result is insensitive to the block size used (Fig. C1) and to the number of artificial ensemble members created (Fig. C2).

*Author contributions.*   ABF and LZ conceptualized the study. ABF established the methodology, applied it to data, and wrote the initial paper draft. ABF and LZ wrote and edited subsequent versions of the paper.

*Competing interests.*   The authors declare that they have no competing interests.





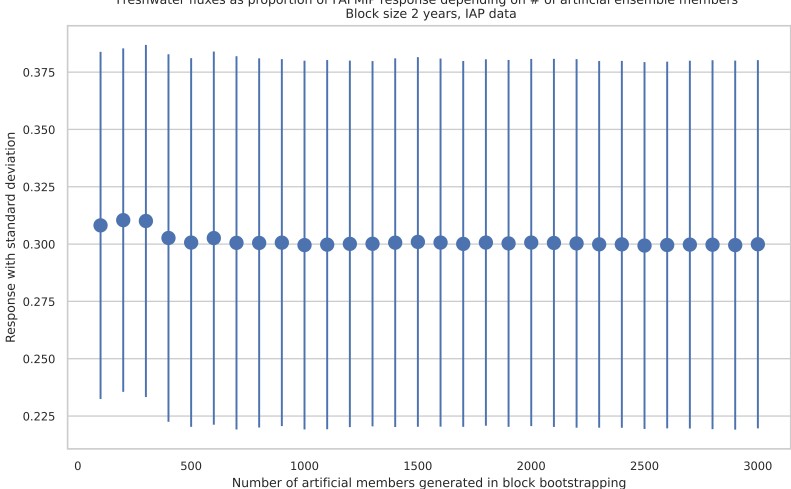

**Figure C2.** The freshwater flux change as a proportion of the FAFMIP perturbation (y axis) versus number of artificial ensemble members (x axis). Here, we applied the method to the IAP data over the period 1975 to 2019 and used a block size of 2 years.

*Acknowledgements.* This work was funded by NSF OCE Grant 2048576 on Collaborative Research: Transient response of regional sea level to Antarctic ice shelf fluxes. We thank Jonathan Gregory and Shafer Smith for the helpful discussions about this work. We also thank Fabrizio Falasca for valuable discussions and feedback on the paper draft.



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
