# Peer review of "Estimating freshwater flux amplification with ocean tracers via linear response theory"

_Earth System Dynamics, 2023_

## Author Comment (AC1)

In the present paper, the authors propose, test and apply a novel methodology to estimate changes in the global hydrological cycle (fresh water fluxes) from trends in ocean salinity. The method based on linear response theory and takes into account regional changes. The authors test their method utilizing ensemble simulation from a climate model (the Community Earth System Model; CESM) and apply it to estimate trends of the hydrological cycle from observations (temperature and salinity from the Institute of Atmospheric Physics). Applying the method to CESM data, show that the proposed methodology can reasonably recover the true freshwater flux of the ensemble mean. The generation of artificial ensembles allow for recovering also changes of individual realizations, i.e. indicate the applicability to observations. However, in this case additional significance criteria for the trend must to be met. Finally, the application to observations give results comparable to previous studies.

As the authors state, estimating the changes in the hydrological cycle as accurately as possible is important and a major challenge. Despite its limitations, response theory can provide an additional method to further improve the estimates.

Overall, I think this is an interesting and valuable study and provides sufficient new and significant information to warrant published. The manuscript is well written and structured and provides a (almost, see below) clear description of the methodology and the results. However, I have a few comments/questions (in random order) the authors may like to consider:

We thank the reviewer for their constructive comments, which we address individually below.

1) Surface temperature as an additional constraint: In lines 121/122 the authors state that they use surface temperature as an additional constraint. I'm wondering what effect this constraint has. How much would the results change with salinity only?

Thank you for the question. The surface temperature is a necessary constraint to help distinguish the response of regional surface salinity to different forcings, which gets blurred after dimensionality reduction. Especially in saltier regions, the response of salinity to heat fluxes and freshwater fluxes is similar, which is likely why the response to those two forcings can't be separated based on surface salinity alone. The additional information from surface temperature constrains the problem by adding enough information that responses of the observables (regional salinity and temperature together) are sufficiently distinct between forcing experiments.

2) Response functions: From Figure 4 (and 5) the salinity responses of the three forcing experiments show some differences. Thus, the response functions (R) obtained from the individual simulations may have different properties as well. How are the actual Rs used for this study are related to the individual once (but, perhaps I missed or overlooked something)? In addition: As stated by the authors, the sixth mixture for the HadOM3 model seem to indicate a non-linear response. Are these data nevertheless contribute to the final Rs?

Thanks for the questions. As you note (from Figures 4 and 5), the responses due to the surface forcings differ between the FAFMIP ocean models that are used to form the response functions.

Here, our goal is to capture the individual response of regional salinity to different forcings when forming $R^h(t)$, $R^w(t)$, and $R^s(t)$ in a way that averages over model dependent responses. Thus, we use $R^h(t)$, $R^w(t)$, and $R^s(t)$ from each ocean model and carry through the procedure to find the response and then take the mean across ocean models at the end. We clarify this in text at line 181 by adding the following sentence: "We set-up Eq. (5) using response functions derived from each FAFMIP ocean model and then take the mean across ocean models (ACCESS-OM2, HadOM3, MITgcm) after solving for the $\frac{dF^h}{dt}$, $\frac{dF^w}{dt}$, and $\frac{dF^s}{dt}$ terms."

As for the HadOM3 model, we do use it despite the non-linear response in the sixth mixture, as we wished to include as many ocean models as possible. However, we performed additional testing which demonstrates that the nonlinearity doesn't impact the results significantly. We solve for the hydrological cycle amplification rate from observations again but ignore the sixth mixture when solving for fluxes from the HadOM3 model (the other two models continue to account for all 6 mixtures). We found a hydrological cycle amplification rate of 4.13 ± 1.21% which is well within the error bounds of the original result reported (4.52 ± 1.21%). The slight difference in results isn't necessarily due to the non-linearity in HadOM3; rather the least squares problem solved is now differently constrained if we ignore the 6th region. However, the similarity in the results builds confidence in the robustness of the reported amplification rate despite the nonlinearity in HadOM3.

3) GMM regions and response functions: As far as I understand, the response functions are derived for the individual GMM regions based on CESM salinity (section 2.2.2). For the observations new GMM regions are defined (Figure 3). However, It seems that the response functions remain the same (based on CESM salinity). Is this the case? If so, how different would be the result when using response functions computed for the observation regions?

Thanks for the comment. For the observations, new response functions are used based on the GMM regions defined. We will clarify this in the text in the observations section. At line 268 in the original manuscript we add this sentence: "Following the steps from Section 3.1, we define the response functions $R^h(t)$, $R^w(t)$, and $R^s(t)$ from the response of salinity and temperature in the FAFMIP experiments in each of these GMM regions."

4) Significance criteria: The authors need to apply additional significance criteria for the trends to capture the true response for individual CESM ensemble members. Unfortunately, the criteria (in my view) are quite subjective (or, better, are fitted to obtain the correct outcome for the given data set). Fortunately(?), the observations met the criteria for all regions. Beside that I'm surprised by this (which, in my view, may indicate important differences between observations and CESM data), I'm wondering how one would proceed in the case where not all regions met the criteria. Or: How large is the contribution of each GMM region to the total response?

Thanks for the comment. We agree that the criteria here are fitted based on testing on the CESM dataset. The motivation was assuming that different ensemble members of the CESM large ensemble sample enough range in the signal to noise ratio (forced response to internal

variability) that we can derive reasonable significance criteria that hold on another dataset. In particular, for these significance criteria to hold across datasets, we needed ensemble members with lower signal to noise ratio than the new data that we wanted to apply the method to; this establishes a "worst case scenario" for which data the method can be applied to.

The fact that the observations easily meet the criteria suggests that observations have a stronger signal to noise ratio than most CESM ensemble members. If it had instead been the case that observations had not met the criteria, we wouldn't have been able to apply the method to observations, as our method could not recover the true forced response due to strong internal variability.

As for the contribution of individual GMM regions, the fitted significance criteria indicate this to some extent: from testing on the CESM ensemble, having strong trends in regions 2 and 6 (which are the regions on average that tend to have the strongest response) is important for whether we can recover fluxes. In other regions, the trend tends to be unclear due to internal variability, but we can still recover fluxes despite this.

5) Effect of the heat flux:  a) It seems that the authors relate changes in heat flux to changes in circulation/transport (e.g. L195) and therefore claim that their method also captures those changes (e.g. L. 332 & abstract). In general this may be true. However, since the heat flux (via the surface temperature) also directly affects evaporation and thus the hydrological cycle, it is not clear to me to what extent the effect of transport changes are really captured (using linear response). The authors may comment on this. In addition, evaporation also enters the heat flux (via the latent heat flux) and I'm wondering whether this matters for the derived (linear) response of salinity to heat fluxes (in particular for regions where evaporation dominates).

Thanks for the questions. Our method assumes that the response of the system (found through ocean salinity and temperature) to each of the FAFMIP forcings is separate. The ocean only FAFMIP experiments are performed by holding other fluxes constant while perturbing just one. For example, in the heat flux experiment, freshwater fluxes and wind stress are held constant while heat fluxes are changed, and so there can be no feedback between the perturbed heat fluxes and the hydrological cycle. Thus, any change in salinity in the FAFMIP heat flux experiment must be due to ocean transport change.

6) Figure 2: The authors may indicate the direction of the flux perturbation (does positive mean into the ocean?)

Thanks for the suggestion. We have added this to the figure caption. Here, positive indeed means downward, into the ocean.

---

## Author Comment (AC2)

In this work, the authors present a means to predict the amplification of the hydrological cycle using observed surface salinity data while accounting for local circulation changes which may obscure the 'true' E-P signal on the ocean's surface. This work represents an important evolution in studies that attempt to infer global water cycle change from ocean salinity observations. Past research has identified an impact of sub-surface ocean warming on ocean circulation, which limits our ability to infer hydrological cycle change from surface properties alone. Three-dimensional analyses, on the other hand, are hampered by the significant uncertainty associated with deep ocean salinity measurements. The authors combine unsupervised learning, idealised FAFMIP experiments, an ensemble of CESM historical simulations and IAP ocean observations, and quantify the regional contribution of ocean circulation changes to salinity change. This estimate thus allows for the production of a new surface salinity-derived estimate of water cycle change which accounts for regional circulation changes.

Linear response theory is the unifying method which draws together these lines of evidence and hinges on two fundamental assumptions – that changes to ocean tracers are a linear sum of freshwater fluxes, heat fluxes and wind stress changes, and that the response of surface tracers to surface forcing is linear. The study does a good job of acknowledging, and where possible, quantifying, the caveats associated with these assumptions.

Overall, I commend the authors on an inspired analytical framework that represents a real step forward in the field. I believe the results are important and should be published. I have some comments, however, on the methodology, which should be addressed prior to publication:

Thank you for the helpful comments and feedback. We address comments individually below.

Comments:

- The choice of input data for the GMM clustering:

It isn't clear to me why the authors used the time-mean salinity distribution in defining the geographical locations of the clusters. Wouldn't it be better to calculate the GMM clusters for each year, then sum them together to get a 'fuzzy' set of boundaries that account for both the probability (which comes out of the GMM method), and the spatio-temporal variability of the salinity field? Perhaps the choice made by the authors wouldn't have a big effect up to 2019 – but surely in the RCP case they would see large changes in the spatial extent of the GMM clusters year-on-year?

Thank you for the suggestion. We agree that it would be interesting to be able to account for how categorization of the GMM would change in time, especially under stronger forcings. However, we think that using a time mean to define the GMM clusters may be advantageous for a few reasons. Firstly, the change in the spatial extent of the GMMs between years would be some combination of forcing and internal variability. We try to smooth over the variability by taking a temporal mean. Secondly, it helps with interpretability to keep defined clusters over time when attributing the change in time series (regional salinity or temperature) to each forcing. If we allowed the clusters to be redefined each year, then the change in salinity in the n-th cluster between two years would both be due to evolving salinity and because the bounds of the region have changed, which obscures the temporal evolution of salinity.

- Choice of number of GMM clusters:

AIC and BIC have different equations, with BIC penalising a larger number of clusters (i.e., a more complex model) more. I found it surprising, then, that BIC and AIC have almost identical profiles in figure S1 for the input data. Can the authors comment on this?

Thank you for the questions. Based on the similarity between the AIC and BIC profiles, we hypothesize that this is because, at the number of mixtures considered, both of the criteria are more determined by the term they have in common (2 ln (L), where L is the maximized value of the likelihood function) rather than the penalization of number of parameters. If we extend out the AIC and BIC plots to consider increased complexity models, the difference in profiles becomes more apparent, as here the penalization may be more important:

[Figure]

Also, typically, the local minimum in AIC/BIC is used as the 'optimal' number of clusters. Where there isn't a local minimum, I would have expected the authors to reach for other quality metrics like the Silhouette Score or the elbow method. This wasn't explored in this work but may give some more insight into the number of clusters chosen. Additionally, where statistical measures like AIC/BIC fail, the authors could have assessed things like RMSE/variance of salinity in each cluster for a variety of cluster numbers, or other physically relevant parameters. Overall, I found the choice of 6 clusters somewhat subjective and would appreciate some more quantitative assessment to back this choice up.

Thank you for the suggestions. The choice of focusing on 6 clusters was due to the slight inconclusiveness of the elbow method. Here, the elbow method on the AIC and BIC implies 5 or 7 clusters for CESM data, and in our view is inconclusive in this range. On the IAP data, the elbow method seems to indicate 5 clusters; however, we again see that this isn't completely clear.

Thanks for your suggestion to test using the silhouette method. We tried using the method and again found that the results were unclear; for example, the mean silhouette score across clusters with cluster numbers ranging from 2 to 8 was very similar (ranging between 0.535 and 0.581). The silhouette score was maximized with cluster numbers 2, 3, or 4 which were the same up to 2 significant digits (0.58). However, based on the elbow method these were not the top contenders for the number of clusters. In fact 2 and 3 clusters would be poor choices based on the elbow method. We also tried another heuristic way of using the silhouette method where we required that no individual cluster had all points below the mean score. However, this metric was true for all clusters when using between 2 and 8 total clusters (the range tested).

As such, in our revised manuscript, we plan on keeping the figures focused on 6 clusters, as a representative number of clusters in a reasonable range indicated by the elbow metric. However, we stress that this is just one example value in a representative range. We have balanced the subjectivity of focusing on 6 clusters by testing the full method on a range of different cluster numbers, where we find there is almost no dependence of the final result on the choice of cluster number (e.g. see Table 2 which shows IAP freshwater flux amplification when running the whole method through with 5, 6, and 7 clusters).

- Plot results on GMM cluster maps

I felt that the results in Figures 4 and 5 were unclear – there is no guarantee that the cluster numbers align with the geographical locations in Figure 3b and d, other than the fact that the mean salinity of each cluster is similar. I think the presentation of these results could be improved by plotting the change in salinity for the FAFMIP experiments onto the cluster locations in lat-lon space. Note that because each cluster has an associated probability, the results won't have the same sharp boundaries as the more deterministic plots in Figure 3.

Thank you for the suggestion. For Figures 4 and 5, the cluster numbers should align with the geographical locations in Figure 3. When using the method to solve for fluxes out of a dataset, we form the response functions using FAFMIP data in the clusters determined for the appropriate dataset. The sharp boundaries of the clusters are created by categorizing each point such that it fully falls into a cluster if the associated probability of being in that mixture is the highest.

We aim to make this clearer in text by editing and adding to sentences at line 188 in the original manuscript: "Here, we validate this hypothesis by examining the effect of the FAFMIP step forcings – heat flux, freshwater flux, and wind stress change – on surface salinity in GMM regions based on CESM data. The CESM regions (Fig. 3a and b) are used to illustrate the FAFMIP response functions here; however, when applying the full flux estimation method on a different dataset of interest, the locations of each response function change slightly according to the GMM fit for that dataset."

Finally, we agree it would be a helpful visualization to show the results of Figure 4 on a map. As we would need 9 subpanels to represent Fig. 4 on a map, we think it may overcomplicate the figure for the main text, but will add it in the supplementary.

In the Abstract, the authors make the point that the sub-tropics host the largest amount of induced ocean circulation change. This is a very important point and represents the first time (in my view) that such a distinction between 'surface flux-induced and 'transport-induced' surface salinity change has been made at the regional scale. I believe the authors could emphasise this distinction more in the Discussion/Conclusions and in Figures 4 and 5, once they have plotted the results onto the clusters in geographical space.

Thanks for the suggestion! We will emphasize the result regarding circulation change in the subtropics.

- Location-based input in GMM

Still on the subject of the GMM clustering, I found it interesting that some clusters (e.g. Cluster 3) cover parts of the Weddell Sea, Labrador Sea, and parts of the sub-tropical gyres in the Pacific. The overall circulation dynamics in these regions is quite different, so it casts some doubt on the ability of the method to distinguish circulation changes in each cluster distinct from surface flux induced salinity changes. It may be, for instance, that temperature-induced circulation changes cancel each other out in all the clusters except cluster 6. Of course, this doesn't negate the global amplification estimate produced, which integrates over all clusters anyway, but I think a major innovation here is the development of regional estimates. I would recommend that the authors try to add location (latitude/longitude) to the GMM input data (properly weighted so as not to overwhelm the salinity input), such that the clusters are less geographically disparate. This will allow for more concrete statements about the regions

experiencing circulation change, as well as a qualitative assessment of the processes that may be contributing to this circulation change.

Thanks for the comment and suggestion. Here, when clustering, we focused just on salinity because this is most in line with the primary goal of estimating fluxes, given the salty gets saltier and fresh gets fresher paradigm. In this context, we commented on the effect of circulation change in the GMM regions from the FAFMIP response functions as it relates to how the salinity signal needs to be partitioned into contributions from fluxes and from circulation change to properly estimate the flux portion.

However, if we were interested in circulation change outside of the flux estimation problem, there's no reason to limit ourselves to only looking at the GMM regions from FAFMIP data, which we used to form the response functions. For example, here we plot the full maps of surface salinity change (last decade minus first decade) from the faf-heat experiment of ocean only FAFMIP (Todd et al. 2020). More in depth analysis of these regional signals would be an interesting follow on study which focuses more on the mechanisms of ocean circulation change and how these imprint on surface salinity.

[Figure]

Change in surface salinity in faf-heat experiments
Mean of last decade minus mean of first decade

- Linear assumption of transient response theory

In Figure 5, I found it somewhat concerning that the cluster where temperature-induced salinity change is greatest (Cluster 6) is also where the linear assumption of the theory breaks down in HadOM3, and qualitatively it looks like the faf-all and linear sum are most different in MITgcm. Is it the case that the greater the temperature-induced circulation change the less we may assume this linear relationship holds? One way to test this could be to make use of a more strongly forced case where all clusters experience significant circulation as well as surface flux-induced change and see if (and for how long) the linear relationship holds. This could be an important result, signalling the validity of this method in the future.

Thank you for the questions and feedback. We agree that it should be better tested how important it is that the cluster with the strongest signal of redistribution (cluster 6) breaks the linearity assumption. To test the sensitivity of the results to this, we performed additional testing where we solve for the hydrological cycle amplification rate from observations again but ignore the sixth mixture when solving for fluxes from the HadOM3 model (the other two models continue to account for all 6 mixtures). Then, we find a hydrological cycle amplification rate of 4.13 ± 1.21% which is well within the error bounds of the original result reported (4.52 ± 1.21%). This builds confidence in the robustness of the reported amplification rate despite the nonlinearity in HadOM3. (See also the response to point 2 of reviewer 1).

The nonlinearity is visible from the FAFMIP response functions which are set at 2xCO2, so we can't exactly test if this nonlinearity would get larger with stronger forcings. However, to give some confidence that this nonlinearity doesn't result in inconsistency when applying our flux estimation method on other data, we can consider results from the application of the method to the CESM ensemble mean data from a later time period (2011 to 2055). We find the response from linear response theory is 0.7544 ± 0.0138 times the FAFMIP forcing. From the target metric (Equation 10 in text) using the true freshwater fluxes, the amplification was equivalent to 0.7339 ± 0.0494. Thus, we find that the method performs well on additional testing over a later time period.

Note that above, the amplification was found as in the original manuscript: block bootstrapping around a linear trend to account for uncertainty and then taking the average and standard deviation of the last 5 years minus the first 5 years. As per your next comment, we will switch to fitting new linear trends to each of the block bootstrapped time series, rather than taking the last 5 years minus the first 5 years. We find that if we assess the trend this way, the response from 2011 to 2055 from our linear response theory method is 0.8303 ± 0.01486, while the target metric from the actual freshwater fluxes is 0.8074 ± 0.0566.

- L222: I notice that the authors take the difference between the first and final 5 years as their change estimate. I would recommend the authors instead take a linear trend over the entire time period and multiply the slope of the trend by the number of years they are interested in to get a change. This would avoid issues associated with aliasing if they catch the model in different phases of decadal (or longer timescale) variability.

Thanks for the comment. As you correctly pointed out, on L222 where we discuss the result of $F^w(t)$ over time from linear response theory applied to the CESM ensemble mean data, we didn't account for variability when reporting the result. Additionally, in the rest of the paper, we account for variability by performing block bootstrapping around the linear trend and then finding the difference between the first 5 years and the last 5 years across all members (mean and standard deviation). To account for your suggestion, we plan on changing this to refitting linear trends to each of the block bootstrapped time series rather than finding the change in each of these time series as the last 5 years minus the first 5 years. This will be applied consistently in a new version of the paper, including here on L222 where we missed accounting for variability. We find that all values go up a little bit (roughly by a factor of 45/40) because the last 5 years minus the first 5 years cuts off 5 years of the trend.

As we will explain in an updated manuscript, we find the mean and standard deviation across the newly fitted linear trends for each block bootstrapped member by using a mixture distribution accounting for each of the newly fitted trends and their uncertainty. This way of accounting for error isn't entirely necessary, but accounts for slightly more error than if we simply take the standard deviation across the change from the newly fit linear trend without accounting for uncertainty in those trends. We summarize the updated results here:

|  | CESM ensemble mean | Observations |
| --- | --- | --- |
| **Target metric from fluxes,** new linear trend metric without accounting for variance of individual newly fit trends | 0.4114 ± 0.0453 | – |
| **Linear response,** new linear trend metric without accounting for variance of individual newly fit trends | 0.3515 ± 0.0344 | 0.3278 ± 0.0647 |
| **Target metric from fluxes,** new linear trend metric *accounting* for variance of newly fit individual trends | 0.4114 ± 0.0602 | – |
| **Linear response,** new linear trend metric *accounting* for variance of newly fit individual trends | 0.3515 ± 0.0432 | 0.3278 ± 0.0828 |

Paragraph beginning L245: Like Reviewer 1, I found the significance criteria here very subjective. This reduces the applicability of the method to other data sets, which may not meet these criteria (though IAP does). Why not use a single signal-to-noise ratio as the cut-off for all clusters/regions?

Thanks for the comment. We agree that the significance criteria used are based on testing on a specific dataset (CESM). These criteria are used because some regions don't have strong trends in any ensemble member. Thus, if we used the strictest signal to noise ratio in all regions, all members are discounted. Yet, we find that even with some regions having insignificant trends, the fluxes can be recovered. Thus, we think that these subjective significance criteria are necessary here but are an area for improvement for future work.

L275: Can you clarify which particular choices the estimate is insensitive to in this sentence? There are a lot of free parameters in the study (I have focussed mostly on GMM), so it would be good to summarise which parameters you have tested sensitivity to.

Thanks for the question. We meant that the estimate was insensitive to the number of mixtures and the block size and number of artificial ensemble members for block bootstrapping. We will reorder and rephrase the surrounding paragraph in an updated manuscript to make this clear.